# Ergonomic Sports Mouthguards: A Narrative Literature Review and Future Perspectives

Talita Suelen De Queiroz [1], Beatriz Serralheiro da Cruz [1], Amir Mohidin Mohamed Demachkia [1], Alexandre Luiz Souto Borges [1], João Paulo Mendes Tribst [2,*] and Tarcisio José de Arruda Paes Junior [1]

1   Department of Dental Materials and Prosthodontics, São Paulo State University (UNESP), São José Dos Campos 12220-000, SP, Brazil; talita.queiroz@unesp.br (T.S.D.Q.); beatriz.serralheiro@unesp.br (B.S.d.C.); a.demachkia@unesp.br (A.M.M.D.); alexandre.borges@unesp.br (A.L.S.B.); tarcisio.paes@unesp.br (T.J.d.A.P.J.)
2   Department of Reconstructive Oral Care, Academic Centre for Dentistry Amsterdam (ACTA), University of Amsterdam and Vrije Universiteit Amsterdam, 1081 LA Amsterdam, The Netherlands
*   Correspondence: j.p.mendes.tribst@acta.nl

**Abstract:** Sports activities can increase participants' risk of orofacial trauma. Therefore, it is crucial to have a comprehensive understanding of various types of sports mouthguards. This study aims to present a narrative literature review on ergonomic mouthguards, including their indications for use, classifications, materials, manufacturing methods, and the current status of additive manufacturing in their production. Research of the existing literature was performed in the Google Scholar, MEDLINE/PubMed, Web of Science, and ScienceDirect databases to identify the currently available publications on the topic of sports mouthguards from January 1951 to August 2023. The terms used were "sports mouthguard", "mouthguards and orofacial traumas", and "mouthguards and additive manufacturing". A total of 920 articles were found, and 39 articles were selected and included in this review. While consensus exists regarding the significance of using sports mouthguards with optimal attributes, there is a need for standardization in the methodology for manufacturing custom-made mouthguards. These can be fashioned with conventional ethylene vinyl acetate (EVA) copolymer reinforcements. Such standardization would ensure uniform stress distribution and guarantee ample protection for oral tissues, allowing for universal reproducibility among dental practitioners. Additive manufacturing stands as an innovative method for fabricating mouthguards, displaying promising benefits. However, the materials and methodologies employed in this workflow still require refinement and characterization for a safe clinical integration.

**Keywords:** sports dentistry; orofacial trauma; sports mouthguard; reinforced mouthguards; 3D printing





## 1. Introduction

Orofacial traumas involve injuries to both soft and hard tissues of the face, such as dislocations, intrusions, extrusions, avulsions, dental fractures, soft tissue lacerations, facial bone traumas, and damage to the temporomandibular joint [1–8]. The occurrence of these injuries is considered a public health issue. Depending on their severity, they can result in a range of consequences and physical limitations for the affected patients [9–11]. Furthermore, in certain cases, the therapeutic process can incur substantial financial costs, and even after treatment, patients may have to cope with permanent sequelae [11–13].

The incidence of these traumas can occur from childhood through adulthood, with a high frequency in sports activities, whether they involve contact or non-contact sports [12,14–18]. With the rise in popularity of contact sports and the encouragement of physical activity participation from a young age, orofacial injuries can be observed [9,10,16,17]. However, many athletes may not be fully aware of the severe implications that can arise if proper preventive measures are not taken [16,18].

The average percentages of athletes in various sports who have experienced any type of orofacial injuries are as follows: wrestling (83.3%), boxing (73.7%), basketball (70.6%), karate (60.0%), jiu-jitsu (41.2%), handball (37.1%), soccer (23.3%), judo (22.3%), and field hockey (11.5%) [8]. The causes can range from direct to indirect traumas. Direct traumas occur when an athlete's facial structures come into contact with another athlete or even equipment during training and competitions [1]. On the other hand, indirect traumas happen when an individual's maxilla and mandible come into intense contact due to falls or impacts, for instance [19].

In cases of significant impacts, the face is the most vulnerable structure in the human body and often the least protected [16]. An athlete has a 10% chance of experiencing an orofacial trauma in each training or competition session, and a 33% to 56% chance over their entire career [20]. However, it is important to note that the prevalence of such injuries varies depending on the type of sport practiced, the athlete's age and gender, and the level of contact in the competition [20,21].

Generally, the frontal region of the maxilla is subjected to horizontal impacts, resulting in a prevalence of 90% of dental injuries occurring in the upper central incisors. These impacts also affect the surrounding structures, causing not only functional and painful issues for the athlete but also aesthetic and psychological concerns [12,14,22,23].

The use of an appropriately designed mouthguard by athletes in sports activities should be encouraged [9,10]. Given the importance of preventing orofacial injuries in sports practice, the American Dental Association recommends the use of mouthguards in various sports, including acrobatic activities, basketball, cycling, boxing, horseback riding, extreme sports, track events, hockey, soccer, gymnastics, handball, ice hockey, skateboarding, lacrosse, martial arts, racquet sports, rugby, skiing, skydiving, softball, squash, surfing, volleyball, water polo, shot putting, weightlifting, and wrestling [11].

Mouthguards are classified into three groups according to their fabrication: stock, boil and bite, and custom-made [1,6,7]. Additionally, they can be fabricated from various materials that can influence their effectiveness [5]. Among the most commonly used materials for producing this device are polyvinyl acetate–polyethylene or ethylene vinyl acetate (EVA) copolymer, polyvinyl chloride, latex, acrylic resin, and polyurethane [2].

Although using a mouthguard in sports activities is essential for preventing or reducing orofacial injuries, impacts affecting the rigid structure while using this device remain significant, especially in the anterior region of the maxilla [6,7]. In this context, studies have been conducted to enhance the effectiveness of mouthguards by incorporating reinforcements, such as laminated layers, air-containing cavities [12], Sorbothane inserts [13], acrylic resin [24], silica mesh [6], titanium [14], sponges, and fiberglass [15]. However, the literature remains inconclusive regarding the best method of reinforcing mouthguards [14,15].

Orofacial trauma resulting from sports activities is common. Therefore, using protective mouthguards is strongly recommended for athletes. Many reports in the literature discuss the effectiveness of different types of sports mouthguards; however, an update about different types of protective mouthguards, materials of fabrication, design, and new technologies used, such as 3D-printed mouthguards, is needed.

Therefore, the present study aims to provide a narrative literature review of the studies conducted on the different kinds of Sports mouthguards, elucidating their indications for use, classifications, materials of fabrication, manufacturing methods, and the current state of additive manufacturing development in the production of these devices.

## 2. Narrative Literature Review Search Strategy

A search for articles on sports mouthguards from January 1951 to August 2023 was performed by two researchers using Google Scholar, MEDLINE/PubMed, Web of Science, and ScienceDirect resources. Only original research and review articles in the English language were included in this review. A total of 920 articles were found and 39 articles were selected and included in this review. The terms used for the search were "sports mouthguard", "mouthguards and orofacial traumas", and "mouthguards and additive

manufacturing". Further searches were conducted in the references list of the relevant articles dealing with the topic of interest. Editorials, Letters to the Editor, Case Reports, and short communication were excluded.

### 2.1. Mouthguards

Given the high incidence of orofacial traumas among athletes during their sports activities, it is essential to encourage the use of effective protection methods [12,23,25]. Mouthguards and facial protectors are two efficient methods for preventing orofacial injuries, with their indications for use varying according to the type of sport being played [17].

Since the first mouthguards were tested in the 1920s, a reduction in oral traumas among boxers was observed, leading to the encouragement of their use among American football athletes [26]. Currently, their significance is well-established in preventing injuries to soft tissues, teeth, and the surrounding bone structures during contact sports [14–16]. It is important to highlight that, in dentistry, other intraoral devices such as occlusal splints or night guards are used for protecting teeth from wear in patients with bruxism, relaxing jaw muscles, and treating temporomandibular disorders [27]. Due to the differences in materials, designs, and indications for use compared to sports mouthguards, these devices will not be addressed within the scope of this review.

In this regard, in 1950, the American Dental Association (ADA) recommended the use of resilient protective devices and mouthguards to prevent or mitigate orofacial injuries, even when the impact is not directly exerted on the tooth [6,7]. After a decade, positive outcomes were observed regarding effectiveness and a reduction in the incidence of damage to orofacial structures, which led to their mandatory use in contact sports [8].

### 2.2. Characteristics and Mechanisms of Protection

The mechanism of protection provided by these devices in reducing orofacial traumas involves absorbing and dissipating the energy from a received impact, thereby preventing it from directly affecting the oral structures. They function somewhat like a cushion, acting as shock absorbers and reducing the severity of trauma to the surrounding structures [8,20,26]. However, for these devices to effectively fulfill their purpose, they rely on various factors including material type, design, correct manufacturing process, appropriate thickness, and sufficient retention [8,22,28].

Ensuring adequate retention of a mouthguard is a crucial step in determining its effectiveness. This is because the device should remain securely in position during sports activities to ensure it effectively fulfills its function in the event of an impact. Moreover, this characteristic can boost athletes' willingness to use the protective device, as it provides comfort without compromising their performance [29].

Another important parameter to consider in the characteristics of a mouthguard is its thickness. It should be sufficient to absorb and dissipate impact energy, providing adequate protection without compromising the athlete's breathing, performance, and comfort. Neglecting these factors can lead to non-compliance with wearing the device [22,28]. While a thicker material tends to absorb more energy [9,28] it will also be more difficult to be used. The mouthguard's thickness needs to fall within the range of 3 to 4 mm to ensure proper impact absorption and comfort [14,22].

In addition to the aforementioned points, a constant analysis of occlusion on the mouthguard's surface is of utmost importance. Using mouthguards with inadequate occlusal adjustments can lead to mandibular fractures upon significant impact or even the development of temporomandibular joint arthritis due to prolonged use. Additionally, wear on occlusal surfaces can occur during use, making periodic checks by a dentist necessary for occlusal adjustments [30].

Several materials have been used for intraoral sports mouthguards, such as latex rubber, vinyl resins, acrylic resins, and acrylic resins combined with silicone. The optimum material for sports mouthguards should exhibit satisfactory mechanical properties with intermediate hardness and adequate energy absorption capacity, derived from good elasticity

and compressive behavior. They should also be biocompatible and comfortable, to allow proper breathing and speech [31]. In this context, ethylene vinyl acetate (EVA) has been considered the most suitable material for mouthguard fabrication due to its satisfactory mechanical properties and ease of manipulation [14,31].

### 2.3. Classification

The ASTM F697-16 standard, titled "Standard Practice for Care and Use of Athletic Mouth Protectors", categorizes the manufacturing and design of mouthguards into three types: (I) thermoplastic type (vacuum-formed and mouth-formed), (II) thermosetting type (mouth-formed), and (III) stock type. However, mouthguards are commonly classified into three categories in the literature and the clinical sports field as the following: stock mouthguards (pre-fabricated), mouth-formed mouthguards (boil and bite), and custom-made mouthguards [24], which will be discussed further below.

#### 2.3.1. Stock Mouthguards (Pre-Fabricated)

Stock mouthguards are defined as plastic trays that cover the teeth. They are the most economical and least precise options. Typically made from polyurethane, polyvinyl chloride, or a copolymer of vinyl acetate and ethylene, and are manufactured in standardized sizes as small, medium, or large. This allows athletes to use them as soon as they acquire them [24]. However, athletes often encounter difficulty in finding information about the thickness of the material, since manufacturers typically just mention small, medium, or large size, but it is evident that there are different thicknesses associated with various available geometries. These mouthguards are characterized by their bulkiness and lack of retention [4]. This is because they are neither custom-made nor adapted for individual patients; instead, users choose them based on their perception of the appropriate size for their dental arch and often without knowledge regarding the different types and required specifications [26]. Users often experience significant discomfort in speech and breathing. Additionally, they must maintain constant occlusion to keep the mouthguard in place, leading to muscular fatigue [4]. Due to their poor retention and inadequate adaptation to the patient's mouth size, this type of mouthguard provides the least protection against orofacial traumas and gives a false sense of security during sports activities [30], as demonstrated in Figure 1. As a result, it is considered contraindicated for use [4], despite being the most financially accessible and easiest to find on the market [26].

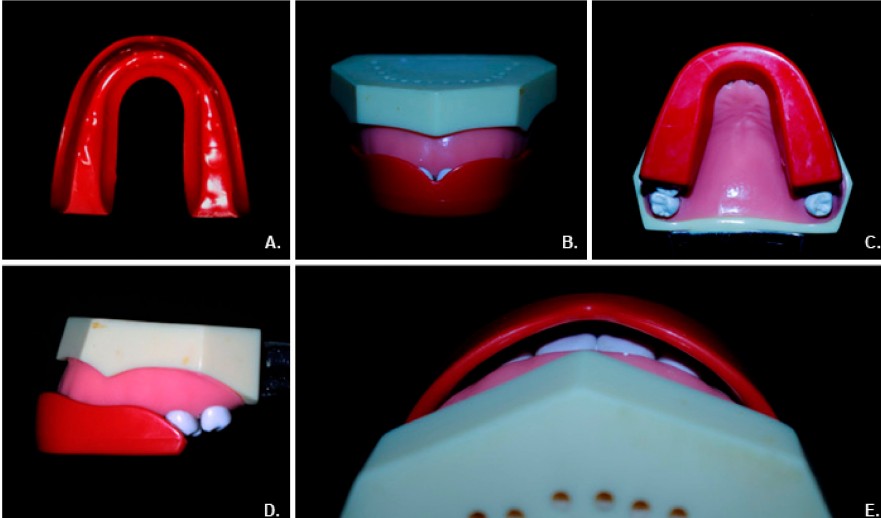

**Figure 1.** Different views of stock mouthguards show the lack of adequate coverage. (**A**) Stock mouthguard. (**B**) Front view showing unprotected dental tissue. (**C**) Occlusal view without occlusion impressions. (**D**) Lateral view showing inadequate coverage to the posterior teeth. (**E**) Space between the anterior teeth and mouthguard indicates inadequate adaptation.

### 2.3.2. Mouth-Formed Mouthguards (Boil and Bite)

This type of mouthguard, typically made of thermoplastic materials, is the most commonly used protective device among athletes [8]. The prescribed procedure for athletes involves immersing these mouthguards in boiling water and softening the material until it becomes malleable. Subsequently, individuals bite, suck, and modify the mouthguard with their tongue and fingers to ensure a proper fit [26], as illustrated in Figure 2. However, this adaptation method is prone to misfits and variations in thickness depending on the force applied to each area by the fingers. Moreover, it may not cover all the necessary regions for protection due to the limited available size options, particularly when it comes to extending coverage to the posterior teeth [4,31]. Moreover, it is important to consider that the forming process of these mouthguards can lead to a reduction in material thickness of around 70 to 99%, depending on the user's technique. This lack of control over the fitting process can compromise the protective capabilities of this type of mouthguard [8].

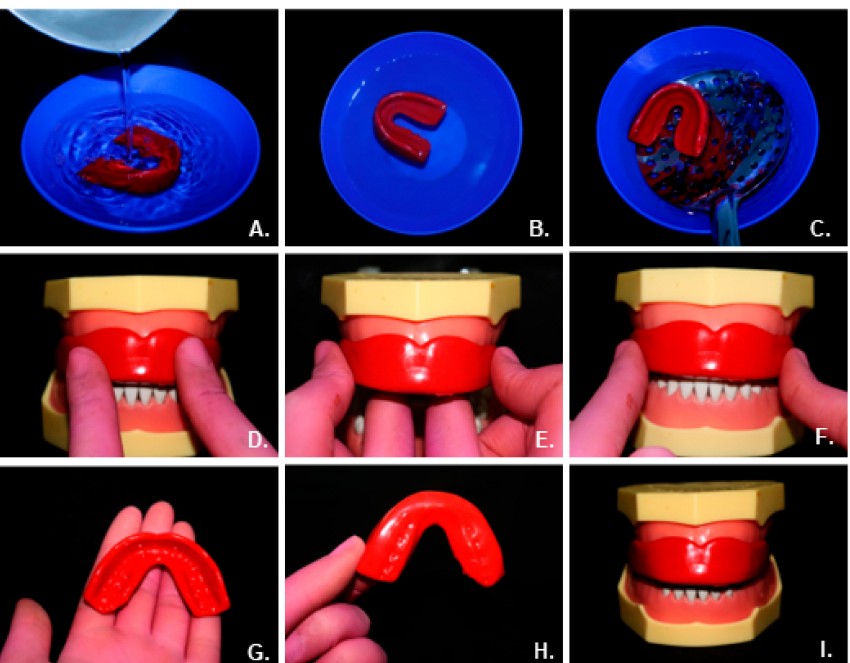

**Figure 2.** (**A**) Boil and bite mouthguard inserted into boiling water. (**B**) Plasticization procedure of the mouthguard in a few minutes. (**C**) Removal process of the mouthguard from boiling water after plasticization. (**D**–**F**) Adaptation procedure of the mouthguard in the mouth. (**G**–**I**) Final result.

### 2.3.3. Ergonomic Custom-Made Mouthguards

Custom-made mouthguards can be categorized as single-layer or laminated. Laminated ones consist of multiple layers of thermoplastic material firmly fused during the manufacturing process [8]. To produce these mouthguards, athletes must visit a dentist to undergo oral impressions and create a plaster model that accurately replicates the patient's mouth structure, including the final molars, labial frenum, palate, and complete vestibular extension. Using this technique, the resulting mouthguard will be precise and efficient with an ergonomic shape. This approach also brings advantages such as enhanced speech capabilities, improved cardiopulmonary function, and reduced discomfort for the athlete [26,31–34].

The production of custom-made mouthguards employs thermoplastic materials that are heated using plasticizing machines and then adapted to the plaster model through vacuum or pressure equipment [33]. The most commonly used material is ethylene vinyl acetate (EVA), which should be heated between 80 to 120 °C [35] and formed to include all teeth in the arch, extend up to 2 mm from the vestibular fornix on the buccal side, and have a 10 mm extension in the palatal region from the gingival margin [8]. After fabrication, the

dentist must perform occlusal adjustments to ensure proper protection for the athlete [30]. The steps of fabrication of custom-made mouthguards are shown in Figure 3.

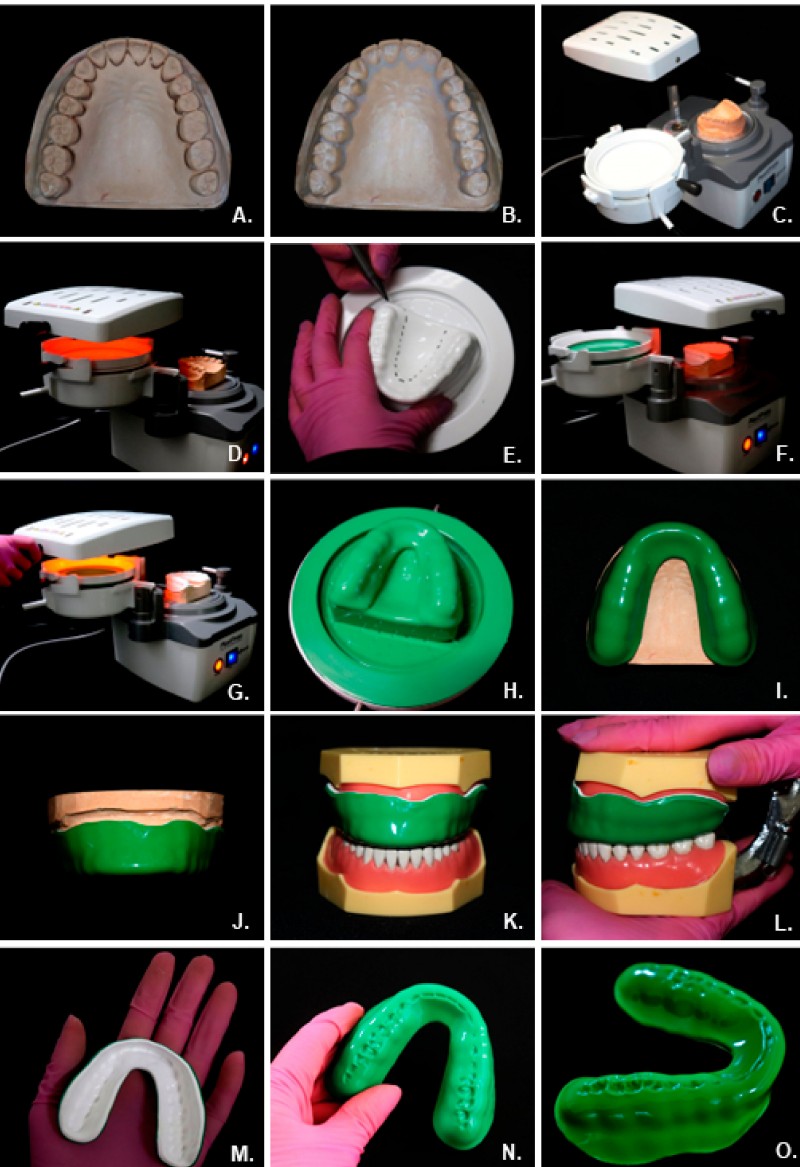

**Figure 3.** (**A**) Obtaining the plaster model. (**B**) Relieving undercut areas with silicone. (**C**) Positioning of the first layer of the EVA 3 mm plate at Plastpress Machine (Bio-art Intelligent Solutions, Sao Carlos, Sao Paulo, Brazil) to perform the vacuum plasticization procedure. (**D**) EVA plate heating for plasticization. (**E**) Delimitation of the mouthguard limit areas. (**F**) Heating the first layer of EVA is already plasticized to allow adhesion with the second. (**G**) Plasticization of the second 3 mm EVA sheet. (**H**) Obtaining EVA plates after the vacuum plasticization process. (**I**) Occlusal view of the final mouthguard. (**J**) Frontal view of the mouthguard on the plaster model. (**K**) Mouthguard on the typodont model. (**L**) The lateral view of the mouthguard shows the correct adaptation. (**M–O**) The final result of the custom-made mouthguard with occlusal registration.

Among all types of mouthguards, the custom-made type emerges as the most suitable choice and offers the highest level of protection against injuries for athletes when properly fabricated [16,33]. This is attributed to its proper fit, retention, comfort, and the absence of a negative impact on performance during sports activities, in comparison to other mouthguards [8].

The principal characteristics and differences between the different kinds of mouthguards can be seen in Table 1.

**Table 1.** Comparative information about different types of sports mouthguards. * The cost is a simplified overview without considering different brands, materials, or manufacturing methods.

| Type of Mouthguard | Level of Protection | Retention/Adaptation | Material | Manufacturing Process | Side Effects | Cost * |
|---|---|---|---|---|---|---|
| Stock Mouthguards (pre-fabricated) | Low | Poor | Polyurethane, polyvinyl chloride, or a copolymer of vinyl acetate and ethylene | Self-use; patients can use them as soon as they acquire them | Discomfort in speech and breathing; muscular fatigue due to the need to maintain constant occlusion | Low (most financially accessible and easiest to find on the market) |
| Mouth-formed mouthguards (boil and bite) | The lack of control over the fitting process can compromise the protective capabilities of this type of mouthguard | Prone to misfits and variations in thickness depending on the force applied to each area by the fingers | Thermoplastic materials | Self-use; patients need to put it in hot water and self-adapt it to their teeth | May not cover all the necessary regions for protection due to the limited availability of size options | Low/Medium |
| Ergonomic custom-made mouthguards | The highest level of protection against injuries for athletes when properly fabricated | The most precise and efficient technique | Thermoplastic materials; the most commonly used material is ethylene vinyl acetate (EVA) | Patients must visit a dentist to undergo oral impressions and create a plaster model that accurately replicates the mouth structure, including the final molars, labial frenum, palate, and complete vestibular extension | Insignificant when compared to the other types of mouthguards | More expensive than the other types of mouthguards, due to the need for professional expertise |

### 2.4. Reinforcements in Mouthguards

As previously mentioned, the use of mouthguards in sports activities is of utmost importance for preventing orofacial injuries. However, even with the use of mouthguards, instances of trauma can still occur [36–38]. This phenomenon arises when the impact force exceeds the protective capacity of the mouthguard, potentially leading to more pronounced injuries, especially in cases of restored or endodontically treated teeth, as these teeth possess altered strength and resilience when compared to intact teeth. Furthermore, in situations involving dental implants, the peri-implant tissues become susceptible to secondary injuries due to the high elastic modulus of the dental substitute [37].

Therefore, numerous studies have been conducted to explore new methods aimed at enhancing the impact absorption capability (effectiveness) of conventional mouthguards against traumatic impacts on anterior teeth [12,14,15]. These studies examined whether the incorporation of intermediate reinforcement layers would enhance the protective response of these devices against impacts, thereby reducing the impact effect on adjacent structures [12]. This entailed designing mouthguards with air cavities and the addition of Sorbothane, metallic wires, sponges, and acrylic inserts [12,14,15], as well as using materials like polyamides, polyethylene terephthalate glycol (PETG), and polyurethanes (TPU) [8].

Reinforcements in EVA layers are feasible due to their long-term adhesive capacity between layers [19,20], market availability, and ease of manipulation [16,19], at ideal temperatures of 80–120 °C [8]. It was observed that EVA mouthguards with a 4 mm air cavity reduced force transmission by 32% compared to conventional mouthguards of the same thickness [12]. Mouthguards with a silicone intermediate layer demonstrated the ability to reduce stress transmission in dental structures by approximately 15–25% when compared to conventional laminated mouthguards [37]. Analysis of mouthguards with sponge inserts between EVA layers demonstrated a 49% increase in impact absorption compared to the control group without reinforcement [35]. Promising results with acrylic reinforcements have also been observed [12]. However, the use of rigid materials does not offer as much clinical safety [35].

Nylon fibers modified with silica are an example of materials that can be used for reinforcing mouthguards during the fabrication process of the device [6,7]. Nylon fibers possess polar amides that enable adjacent chains to form hydrogen bonds, which enhances their crystallinity, strength, and durability. The effectiveness of reinforcements using these fibers depends on their structural design and orientation, as well as the applied force and their interaction with the material to be reinforced for load transfer [39]. Moreover, one of the most successful methods to enhance the mechanical properties of acrylic resin is the incorporation of silica nanoparticles, which have significant effects on both mechanical and thermal aspects. Based on this, studies have been conducted where this structure was incorporated into polymers like acrylic resin and bisacrylic to fabricate temporary fixed prosthetic structures [39] and mucosa-supported and implant-supported complete arch dentures [39–41], resulting in improvements in the strength of the tested structures. With these positive results, a proposal was made by our research group to increase the absorption of impacts using this fiber as a reinforcement in customized mouthguards; this proposal is under development and an example is shown in Figure 4 below.

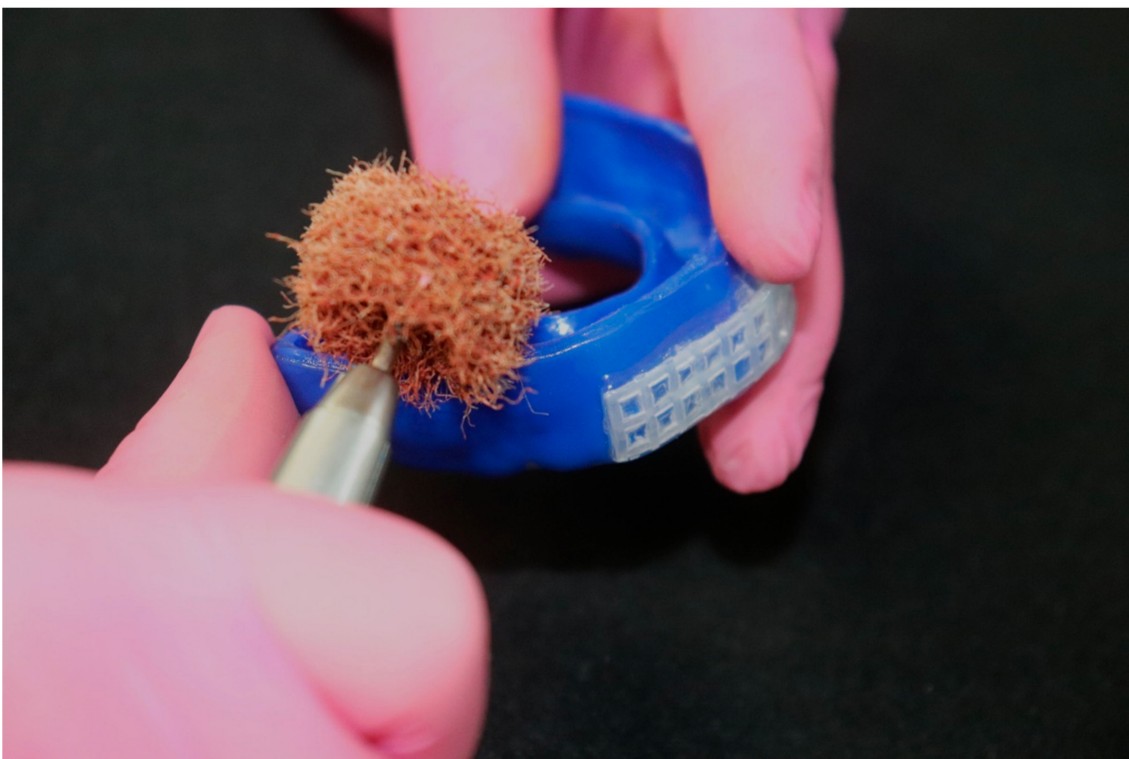

**Figure 4.** Reinforced mouthguard finishing procedure.

### 2.5. Additive Manufacturing

With the rise of digital dentistry and the advancement of CAD/CAM (computer-aided design/computer-aided manufacturing) technology, additive manufacturing has been proposed to produce mouthguards. This is due to the incorporation of new materials and improvements in their replication, as well as the ability to provide individualized solutions with greater accuracy, coupled with the advancement of technical knowledge in this field [8,42,43].

This approach serves to reduce the drawbacks associated with conventional techniques used to create EVA mouthguards, which often involve pressure and vacuum molding. These traditional methods can lead to drawbacks such as reduced final thickness and delamination, especially when incorporating multiple layers of EVA during the molding process [42].

Although additive manufacturing offers innovative solutions for fabricating mouthguards, it is essential to evaluate whether the available materials exhibit mechanical performance comparable to conventional mouthguards made from EVA. This evaluation pertains to factors such as stress absorption, stress distribution, tensile strength, flexural strength, toughness, hardness, fluid absorption, material solubility, and biocompatibility [42,44]. However, only a limited number of studies in the literature have addressed the fabrication of mouthguards using additive manufacturing techniques [8]. Existing studies indicate that there are currently no discernible benefits in terms of the mechanical behavior of the available materials for 3D-printed mouthguards that surpass the performance of EVA [8,42]. Nevertheless, it is worth noting that the impact testing methodologies used are still simplified [42]. Moreover, the market offers various polymer types that require testing due to their distinct compositions. Additive manufacturing presents advantages in terms of customization and control over the final device, distinguishing it from conventional methods [45].

Given the advantages of additive manufacturing in producing intraoral mouthguards, it becomes evident that this technique can offer favorable retention, appearance, and comfort, producing outcomes similar to conventionally fabricated custom-made mouthguards.

However, when considering occlusal balance, it has been observed that 3D-printed mouthguards exhibit balanced occlusal contacts, providing a symmetrical bilateral occlusion in centric occlusion. This is in contrast to conventionally fabricated mouthguards, which may show variations in thickness [45]. This characteristic enables proper stability of the device within the intraoral environment and facilitates better distribution of absorbed forces during traumatic impacts. Figure 5 shows the steps for fabrication of a 3D-printed mouthguard with Dima print mouthguard resin (Kulzer & Co. GmbH, Wehrheim, Germany).

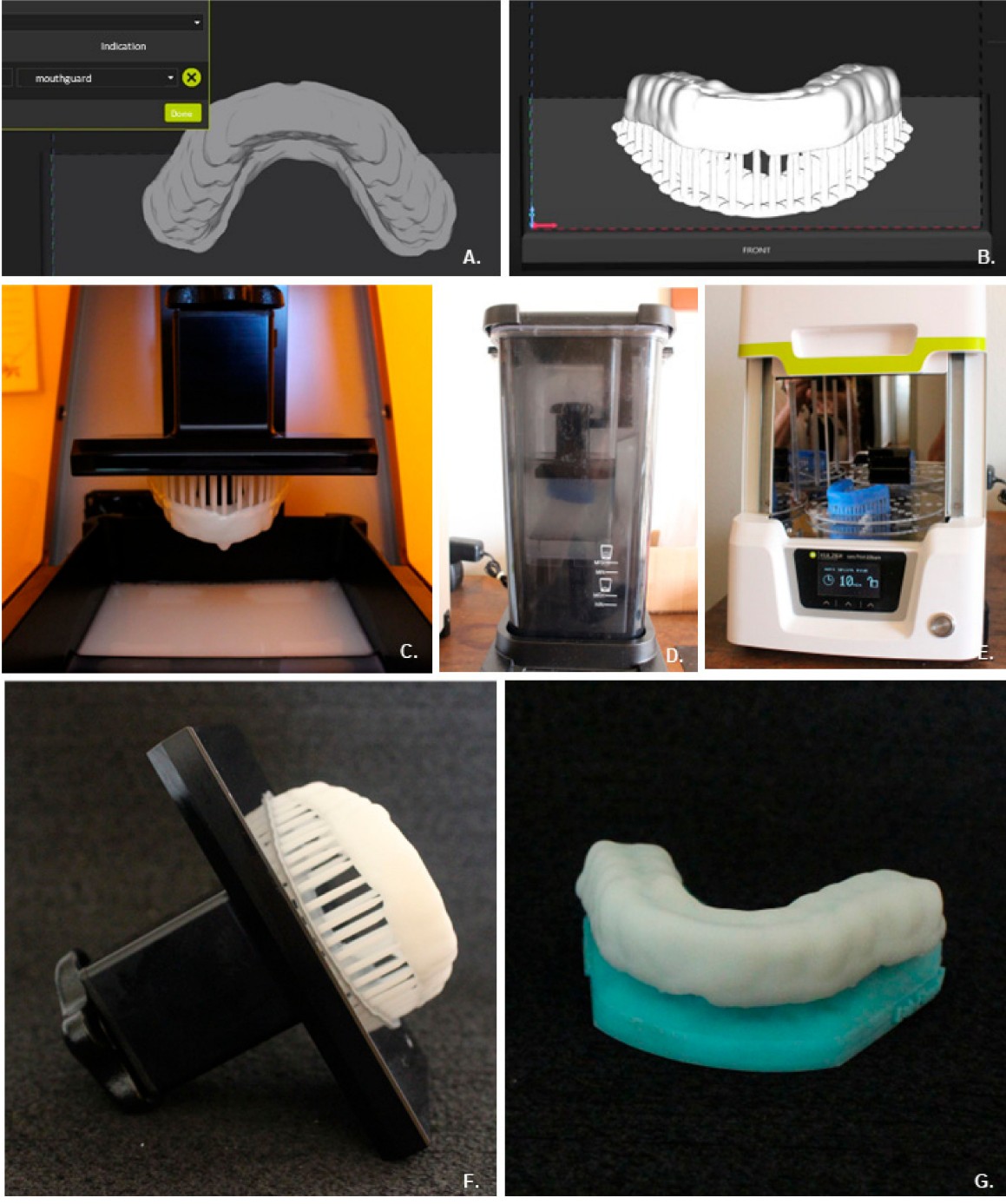

**Figure 5.** Steps for fabrication of a 3D-printed mouthguard. (**A**) Imported file after individualized case planning. (**B**) File placement for 3D printing. (**C**) The 3D printing process in Dima print mouthguard resin using a Cara 3D printer (Kulzer & Co. GmbH, Wehrheim, Germany), following manufacturer's recommendations. (**D**) Washing process in isopropyl alcohol following printing. (**E**) Cure processing. (**F**,**G**) The mouthguard after completing all manufacturing procedures.

Additive manufacturing has displayed such promising growth potential that has spurred the development of intelligent printed materials, known as 3D-printed smart materials. These materials possess the capability to adapt to external conditions such as pH, humidity, and temperature by utilizing their shape memory effect. This advancement is considered significant progress in the field and opens up avenues for further research in this area, with the potential for future clinical implementation [43].

There is still no consensus in the literature regarding materials that can be considered safe as alternatives for EVA in the production of mouthguards. This necessitates further studies that analyze various types of polymers, their characteristics, and their behavior under impact tests with well-defined methodologies. However, additive manufacturing has demonstrated substantial potential for success in the future.

## 3. Discussion

Orofacial injuries encompass a range of traumas, including dental crown fractures, soft tissue lacerations to the tongue and gums, facial bone traumas, temporomandibular joint damage [2,6–8,11], root fractures, concussions, subluxations, extrusions, intrusions, dislocations, and dental avulsions [20]. Such traumas in the oral and maxillofacial region can lead to inconveniences, causing a decline in athletes' performance and potential withdrawal from important competitions. Beyond this, these injuries can alter facial expressions, hinder daily actions like chewing and speaking, and negatively influence nutrition. As a result, they can trigger a cascade of effects, including muscle mass loss, reduced overall performance, diminished strength, and disrupted game rhythm. Moreover, the consequences of these changes are not limited to the athletes themselves. Clubs, agents, and sponsors invested in these athletes are also affected, as these injuries can lead to significant financial troubles. In severe cases, athletes may experience psychological implications due to the permanent consequences [20,25,46].

Currently, the ADA recommends the use of mouthguards for 29 different sports disciplines [2,8]. These devices play a crucial role in reducing orofacial injuries by preventing tooth fractures or displacement [34], protecting against jaw fractures, minimizing the risk of soft tissue lacerations through teeth and soft tissue separation, and decreasing the likelihood of concussions by stabilizing the head through mouthguard use [2]. Adequate coverage provided by EVA material across all teeth and structures enables the effective dissipation of forces. Moreover, impact reduction for the temporomandibular joint complex is achieved through the space created between the condyle head and glenoid fossa, resulting from teeth separation via the mouthguard [4].

Although mouthguards effectively lower the risk of orofacial injuries, some athletes encounter difficulties when using them. They report issues such as instability, a sensation of dry mouth, bad breath, difficulty in speaking and breathing, nausea, and a perceived decline in performance [34]. However, a systematic review conducted by Caneppele and colleagues in 2017 demonstrated that custom-made mouthguards presented the least interference in speech, breathing, and the sensation of dry mouth among the various types of mouthguards. Furthermore, these mouthguards showed better adaptation to the oral cavity and lower incidence of nausea among the athletes analyzed. Custom-made mouthguards provide results closer to what is considered adequate, offering protection to orofacial structures during sports activities [47,48].

Usually, mouthguards are fabricated for the maxillary arch because it is more prominent and prone to a higher incidence of trauma, especially the upper central incisors. Nevertheless, stock mouthguards that cover both the maxillary and mandibular arches at the same time are available; however, they have the disadvantages of stock mouthguards in terms of adaptation and size. Additionally, they are bulky, which can interfere with speech and make them difficult to tolerate [26].

Among the materials studied to fabricate mouthguards, EVA is the most commonly utilized material, due to its ability to provide suitable physical properties to the device, enabling impact absorption and distribution throughout its structure, thereby reducing the

incidence of stress on relevant adjacent structures [6,7,24]. Additionally, it is easy to handle, exhibits minimal water absorption, and allows the incorporation of reinforcement layers within the mouthguard [6,7].

Given the biomechanical behavior of the different types of mouthguards available, the use of ergonomic custom-made devices is recommended for sports that involve high speeds, heights, and physical contact, as well as other sports that can cause orofacial trauma during practice, for both amateur and high-performance athletes, since the stock (pre-fabricated) and mouth-formed (boil and bite) devices do not meet the requirements to guarantee adequate retention [26]. The suggested thickness for creating EVA mouthguards to ensure effective impact absorption capability ranges between three to four millimeters [14]. Westerman and colleagues (2002) found that increasing the thickness beyond 4 mm resulted in only a minor enhancement in impact absorption. However, this slight improvement would not offer significant benefits and could potentially compromise user comfort, thereby potentially discouraging the usage of the device [2].

The incorporation of reinforcements into EVA mouthguards reveals significant improvements in the mechanical performance of these devices, such as enhanced impact absorption compared to conventional mouthguards. Various forms of reinforcements have been studied, and their effectiveness has been proven through in vitro studies. These forms include the inclusion of an air cavity in 4 mm of EVA [12,48], the addition of an intermediate layer of Sorbothane as reinforcement, reducing the transmission of impact to adjacent structures by approximately 30% [36], the insertion of a sponge between two layers of EVA [12], and the utilization of a silicone intermediate layer [38]. While these studies present favorable outcomes for reinforcement incorporation, it is important to note that the methodologies employed in these studies have limitations in terms of applied load, with lower impact magnitude compared to real-life situations faced by athletes. It is essential to acknowledge the difficulty in standardizing load application in vitro studies, but it is worth developing methodologies that involve more realistic tests and analyses aimed at understanding the behavior of adjacent structures when subjected to impacts using different types of mouthguards [48]. Additionally, a combination of two methodologies, such as in vitro and in silico, could provide validation for the studied methodologies [49,50].

Athletes consider two factors when choosing intraoral protection devices: the cost and the time required for acquisition. Stock mouthguards are the most cost-effective, followed by 'boil-and-bite' mouthguards, which are readily available in sports stores and pharmacies, allowing athletes to make their own choices based on what they believe is suitable. Despite the issues associated with these types, such as lower protection rates compared to custom-made mouthguards, they offer quick acquisition without the need for multiple appointments. In contrast, custom-made mouthguards require several appointments that include impression taking, device adjustment appointments, and follow-ups. While this individualized approach is advantageous in terms of protection, some patients view its drawbacks of higher time and cost as not worth the trade-off. However, addressing this concern through educational efforts, emphasizing differences in protection levels and potential trauma-related issues can encourage athletes to choose custom-made mouthguards.

Regarding the potential for the use of additive manufacturing in mouthguard fabrications, mouthguards using Agilus30 resin did not show superior performance when compared to conventional EVA mouthguards under free-fall weight impact tests. However, conventional mouthguards exhibited surface imperfections such as the presence of microbubbles and variations in thickness [42]. In contrast, the 3D-printed mouthguards displayed surface uniformity as well as consistent thickness, suggesting the need for further research analyzing different polymer types to establish a comprehensive recommendation for the adoption of printed materials as a substitute for conventional EVA mouthguards. Additionally, there is a need for better-designed impact tests employing more realistic models and controlled force applications, as well as assessments of proposed polymer characteristics. While a definitive recommendation for the safe clinical implementation of printed materials is not yet established, the occlusion achieved through 3D-printed

mouthguards is superior to conventionally thermoformed mouthguards. The conventional mouthguards exhibited premature contacts in the second molar region, whereas the 3D-printed mouthguards presented a satisfactory bilateral balanced occlusion, minimizing significant adjustments that could reduce the final device thickness [45].

According to the aforementioned, there is a need to develop mouthguards with higher stress-absorption efficiency. It is crucial to explore the possibility of using different reinforcing materials and new fabrication methodologies for these devices, such as additive manufacturing. In terms of relevant research methods, in silico and in vitro approaches are valuable for evaluating the varying protective capacities among available and developed mouthguards [51–54].

Moreover, there exists a vast range of research opportunities in the field of mouthguards, particularly regarding force dissipation, reinforcement incorporation, and additive manufacturing. Given the pivotal importance, it is imperative to undertake studies that strive to elevate the stress-absorption capacity of these devices. This necessitates the exploration of biocompatible materials that not only offer structural reinforcement but also ensure athletes' comfort during prolonged use.

## 4. Final Considerations

Currently, there is a consensus on the importance of using sports mouthguards fabricated according to ideal ergonomic characteristics. However, standardizing the manufacturing method for creating custom-made mouthguards and reinforced materials is essential. The aim is to ensure proper stress distribution and adequate protection of oral tissues, enabling universal adoption among dental practitioners. Additive manufacturing presents an innovative approach to fabricating mouthguards with promising benefits. Nonetheless, the polymeric materials and manufacturing processes need further refinement to ensure the safe clinical integration of these devices.

**Author Contributions:** Conceptualization, T.S.D.Q. and J.P.M.T.; Methodology, T.S.D.Q. and B.S.d.C.; Validation, A.M.M.D., A.L.S.B. and J.P.M.T.; Formal Analysis, T.S.D.Q. and A.M.M.D.; Investigation, T.S.D.Q., B.S.d.C. and A.M.M.D.; Data Curation, T.S.D.Q. and A.M.M.D.; Writing—Original Draft Preparation, T.S.D.Q.; Writing—Review and Editing, T.S.D.Q., B.S.d.C., A.M.M.D., A.L.S.B., J.P.M.T. and T.J.d.A.P.J.; Visualization, T.S.D.Q. and A.M.M.D.; Supervision, A.L.S.B., J.P.M.T. and T.J.d.A.P.J.; Project Administration, J.P.M.T. and T.J.d.A.P.J.; Funding Acquisition, A.L.S.B., J.P.M.T. and T.J.d.A.P.J. All authors have read and agreed to the published version of the manuscript.

**Funding:** This research was funded by the São Paulo Research Foundation (FAPESP), grant numbers 22/11307-9 and 21/11159-7.

**Institutional Review Board Statement:** Not applicable.

**Informed Consent Statement:** Not applicable.

**Data Availability Statement:** All relevant data are available within the manuscript.

**Acknowledgments:** The authors acknowledge Kulzer (Kulzer South America Ltda., Água Branca, São Paulo, Brazil) for providing the didactic material for Figure 5.

**Conflicts of Interest:** The authors declare no conflict of interest.

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
