# Peer review of "Ergonomic Sports Mouthguards: A Narrative Literature Review and Future Perspectives"

_applsci, doi:10.3390/app132011353_

Round 1
Reviewer 1 Report
Dear authors, thank you very much for this paper, focusing on a very interesting and developing topic in dentistry. Dentistry in sports is an upcoming field in our profession.
However, i think corrections or a exact focus of your paper are necessary.
Title: Is your paper really a review or a descriptive paper of possible and available devices. Please clarify and adjust. Reorganize the paper. Therefore a review is difficult in this stage.
Abstract: There is no information about the review process. Which databases did you use, which search strategy, inclusion, exclusion criteria?
Introduction: which orofacial injuries? Please clarify the effect of mouthguards. Please give more information about the needs of our patients when choosing a mouthguard.
Please give a clear aim of your paper? What is the real purpose of the paper? Mouthguards or injuries in sports?
2. Literature Review:
Please provide the search strategy, the search strings and used databases. These points are missing. The search strategy should fit the aim of your paper. Please give information about the inclusion and exclusion criteria. Tables and figures might help.
2.1
Actually 2.1 should be included in the introduction section. The aim of the paper are the different mouthguards.
Please re-organize the entire paper for resubmission.
Some minor corrections are needed.
Author Response
Dear authors, thank you very much for this paper, focusing on a very interesting and developing topic in dentistry. Dentistry in sports is an upcoming field in our profession.
However, i think corrections or a exact focus of your paper are necessary.
Dear reviewer, thank you for your time and effort in evaluating this study. The corrections have been addressed and the point-by-point explanation is presented below:
Title: Is your paper really a review or a descriptive paper of possible and available devices. Please clarify and adjust. Reorganize the paper. Therefore a review is difficult in this stage.
Thank you for pointing this out. It is a narrative review and we have accordingly added type of this review to the title.
Abstract: There is no information about the review process. Which databases did you use, which search strategy, inclusion, exclusion criteria?
We agree with you. Therefore, we have added detailed information about the review process that we did in the abstract and under the heading “Narrative Literature Review Search Strategy”. (line 86)
Introduction: which orofacial injuries? Please clarify the effect of mouthguards. Please give more information about the needs of our patients when choosing a mouthguard.
Thank you very much for raising these points. Regarding orofacial injuries, our specific focus is on sports-related orofacial injuries. Under the subheading "Sport-Related Orofacial Traumas," we have discussed various types of traumas associated with sports, as well as their consequences and impacts on athletes.
(line 73)
“Orofacial traumas include injuries in soft and hard tissues of the face, such as dis-locations, intrusions, extrusions, avulsions, dental fractures, soft tissue lacerations, facial bone traumas, and damage to the temporomandibular joint [17] [3] [8]. The occurrence of these injuries is considered a public health issue. Depending on their severity, they can result in a range of consequences and physical limitations for the affected patients [18] [19]. Furthermore, in certain cases, the therapeutic process involves substantial financial costs, and even after treatment, patients may have to deal with permanent sequelae [8].”
(line 114)
Numerous consequences can arise for an individual who experiences orofacial injuries. These include abnormal exfoliation of primary teeth, disruptions in the eruption of permanent teeth in the case of children, tooth discoloration following trauma, the occurrence of dental abscesses, and even tooth loss. Additionally, depending on the impact, adjacent bone structures can be affected, leading to the possibility of fractures. In more severe cases, the athlete can experience cervical and brain injuries [19]. These factors can impact the individual's daily life, resulting in limitations in speech, facial expressions, eating, and even nutritional and psychological damage. As a result, the athlete's performance recovery can be severely hindered, restricting their participation in competitions [25]. Hence, the involvement of dentists in injury prevention is extremely important [17] [18].”
Regarding the effect of the mouthguard, under the subheading titled 'Characteristics and Mechanical Properties,' there is a discussion about the effect of mouthguards and their protection mechanism. Therefore, we have revised the subheading title to 'Characteristics and Protection Mechanism.'
Please give a clear aim of your paper. What is the real purpose of the paper? Mouthguards or injuries in sports?
The purpose of this review is to give an update about for the readers about the different types of protective mouthguards, materials of fabrication, design, and new technologies used, such as 3D printed mouthguards. Accordingly, we have added a paragraph (line 70) mentioning the aim of this study.
“Orofacial trauma resulting from sports activities is common. Therefore, using protective mouthguards is strongly recommended for athletes. Many articles in the literature discuss the effectiveness of different types of sports mouthguards; however, an update about different types of protective mouthguards, materials of fabrication, design, and new technologies used, such as 3D printed mouthguards, is needed.
Thus, the present study aims to provide a narrative literature review of the studies conducted on the different kinds of Sports mouthguards, elucidating their indications, classifications, materials of fabrication, manufacturing methods, and the current state of additive manufacturing development in the production of these devices.”
- Literature Review:
Please provide the search strategy, the search strings and used databases. These points are missing. The search strategy should fit the aim of your paper. Please give information about the inclusion and exclusion criteria. Tables and figures might help.
Thank you for raising this point out., we have added detailed information about the review process that we did in the abstract and under the heading “Narrative Literature review Search Strategy”. (line 86)
2.1
Actually 2.1 should be included in the introduction section. The aim of the paper are the different mouthguards.
Please re-organize the entire paper for resubmission.
We have merged 2.1 in the introduction and reorganized the entire paper.
Reviewer 2 Report
Dear authors, thank you for submitting the manuscript “Ergonomic sports mouthguards: literature review and future perspective”.
I have read your manuscript and here is my feedback:
Please mention in your title the type of literature review (narrative, scoping, systematic, etc).
You mentioned the classification of mouth guards according to their fabrication, and materials.
Please briefly describe the differences regarding mouth guards for sports and night guard for bruxism.
Please expand further if there is a classification of fabrication/designs related to the type of sport (example if there are differences in design).
The section 2. Literature Review is very weak and lacks important information such as what was your criteria, inclusive and exclusive search. How many articles you excluded and how many articles ended up in the review.
Please inform what searching data you used.
Please inform who performed the review (a single person or a group).
For the stock mouth guard, please mention if companies provide different thicknesses of the material.
You showed images for stock, boil and bite, and customized mouth guards. If possible, please show images for the “Additive Manufacturing” type of mouth guard that you describe in the 2.7 section.
You are only displaying maxillary mouth guards, please mention reasons or clinical scenarios where mandibular mouth guard could be an option.
It would also be interesting to mention about cost differences among the types of mouth guards that you describe in the manuscript.
Please provide information regarding the time needed (appointments) for the fabrication, that could potentially lead patients to select one of the types of mouth guards.
Finish the author contribution and replace the X.X. and X.Y. by authors names.
Minor English revision is required.
Author Response
Dear authors, thank you for submitting the manuscript “Ergonomic sports mouthguards: literature review and future perspective”.
I have read your manuscript and here is my feedback:
Please mention in your title the type of literature review (narrative, scoping, systematic, etc).
Thank you for the suggestion. We have accordingly added the type of this review to the title. You mentioned the classification of mouthguards according to their fabrication, and materials.
Please briefly describe the differences regarding mouth guards for sports and night guard for bruxism.
Thank you for pointing this out. We have added a paragraph regarding this point. “Line 158”
“It is important to highlight that in dentistry, there are other intraoral devices such as occlusal splints or night guards that are used for protecting teeth from wear in patients with bruxism, relaxing jaw muscles, and treating temporomandibular disorders [27]. Due to the differences in materials, designs, and indications for use compared to sports mouthguards, these devices will not be addressed within the scope of this review.
Please expand further if there is a classification of fabrication/designs related to the type of sport (example if there are differences in design).
Thank you for the suggestion. We have added accordingly the following “In view of the biomechanical behavior of the different types of mouthguards available, the use of ergonomic custom-made devices are recommended for sports that involve high speeds, heights, and physical contact, as well as other sports that can cause orofacial trauma during practice, for both amateur and high-performance athletes, since the stock (pre-fabricated) and mouth formed (boil and bite) devices do not provide the necessary requirements to guarantee adequate retention [26].” (line 508
The section 2. Literature Review is very weak and lacks important information such as what was your criteria, inclusive and exclusive search. How many articles you excluded and how many articles ended up in the review.
Thank you very much for raising these points. Therefore, we have added detailed information about the review process that we did in the abstract and under the heading “Narrative Literature review Search Strategy” (line 86)
Narrative Literature Review Search Strategy
Articles on Sports Mouthguards were searched from January 1951 to august 2023 was performed by one person using Google Scholar, MEDLINE/PubMed, Web of Science, and ScienceDirect resources. Original research and review articles in the English language were only included in this review. A total of 920 articles were found and 39 articles were selected and included in this review. The terms used for the search were “sport mouthguard”, “mouthguards and orofacial traumas”, “mouthguards and additive manufacturing”. Further searches were conducted in the references list of the relevant articles dealing with the topic of interest. Editorials, Letters to the Editor, Case Reports and short communication were excluded
Please inform what searching data you used.
Google Scholar, MEDLINE/PubMed, Web of Science, and ScienceDirect
Please inform who performed the review (a single person or a group).
The review was performed by two persons.
For the stock mouth guard, please mention if companies provide different thicknesses of the material.
Thank you for raising this point. Usually, companies do not provide information about the thickness of the stock mouthguard. However, the companies provide information about mouthguard sizes as small, medium, and large. We have added this point in the manuscript (line 271)
“However, athletes often encounter difficulty in finding information about the thickness of the material, since manufacturers typically just mention small, medium or large size, but it is evident that there are different thicknesses associated with various available geometries.”
You showed images for stock, boil and bite, and customized mouth guards. If possible, please show images for the “Additive Manufacturing” type of mouth guard that you describe in the 2.7 section.
We have added images for mouthguards fabricated by 3D printed technology. (line 435)
You are only displaying maxillary mouth guards, please mention reasons or clinical scenarios where mandibular mouth guard could be an option.
Because, according to the literature, the frontal region of the maxilla is subjected to horizontal impacts, resulting in a prevalence of 90% of dental injuries occurring in the upper central incisors. (line 64)
We have also added a paragraph regarding this point in the discussion (Line 489)
“Usually, mouthguards are fabricated for the maxillary arch because it is more prominent and prone to a higher incidence of trauma, especially the upper central incisors. Nevertheless, stock mouthguards that cover both the maxillary and mandibular arches at the same time are available; however, they have the disadvantages of stock mouthguards in terms of adaptation and size. Additionally, they are bulky, which can interfere with speech and make them difficult to tolerate [26].”
It would also be interesting to mention about cost differences among the types of mouth guards that you describe in the manuscript.
Thank you for pointing this point out. We have added a new table “table 1” that summarize the different characteristics of different types of mouthguards including the differences in the cost among them.
Please provide information regarding the time needed (appointments) for the fabrication, that could potentially lead patients to select one of the types of mouth guards.
We have added information about this point in Table 1 and in the discussion (line 536)
“Athletes consider two factors when choosing intraoral protection devices: cost and the time required for acquisition. Stock mouthguards are the most cost-effective, followed by 'boil-and-bite' mouthguards, which are readily available in sports stores and pharmacies, allowing athletes to make their own choice based on what they believe is suitable. Despite the issues associated with these types, such as lower protection rates compared to custom-made mouthguards, they offer quick acquisition without the need for multiple appointments. In contrast, custom-made mouthguards require several appointments that include impression taking, device adjustment appointments, and follow-ups. While this individualized approach is advantageous in terms of protection, some patients view it as a drawback due to the time-consuming and higher cost. However, addressing this concern through educational efforts, emphasizing differences in protection levels and potential trauma-related issues can encourage athletes to choose custom-made mouthguards.
Finish the author contribution and replace the X.X. and X.Y. by authors names.
We have added the author's contribution.
Reviewer 3 Report
The reviewer really appreciates the efforts of the authors to conduct this review which has good clinical significance. The manuscript is well written. There is no major issue in the manuscript. However, there are a few recommendations from the reviewer side. The reviewer believes this will improve the quality of the manuscript and make it interesting for the readers
It would be nice if the author could add a paragraph before the objective stating the problem statement and rationale of the current review. For example, there are several articles on this topic, however, they are missing some aspects or there might be several recent innovations in this area that need an updated review……..
The author can add descriptive and comparative information about the materials used to prepare mouthguards.
Good luck
Author Response
The reviewer really appreciates the efforts of the authors to conduct this review which has good clinical significance. The manuscript is well written. There is no major issue in the manuscript. However, there are a few recommendations from the reviewer side. The reviewer believes this will improve the quality of the manuscript and make it interesting for the readers.
It would be nice if the author could add a paragraph before the objective stating the problem statement and rationale of the current review. For example, there are several articles on this topic, however, they are missing some aspects or there might be several recent innovations in this area that need an updated review……..
Thank you for pointing this point out. We have added a paragraph according to this suggestion. (line 104)
“Orofacial trauma resulting from sports activities is common. Therefore, using protective mouthguards is strongly recommended for athletes. Many articles in the literature discussed the effectiveness of different types of sports mouthguards; however, an update about different types of protective mouthguards, materials of fabrication, design, and new technologies used, such as 3D printed mouthguards, is needed.
Thus, the present study aims to provide a narrative literature review of the studies conducted on the different kinds of Sports mouthguards, elucidating their indications, classifications, materials of fabrication, manufacturing methods, and the current state of additive manufacturing development in the production of these devices.”
The author can add descriptive and comparative information about the materials used to prepare mouthguards.
We have added a table demonstrating the different characteristics of different types of mouthguards including the differences in the cost among them (table 1).
Good luck
Thank you.
Round 2
Reviewer 1 Report
Dear Authors
Thank you for this new version of your work. I think the paper is improved. However, there are some corrections still needed.
The title is improved. Furthermore, the abstract is now in an acceptable form.
The introduction is improved. Thank you.
In the material and methods section: Why did you not use "dental traumatology" or "tooth fractures" as search string. This might lead to a higher number of possible papers for inclusion. Please explain and discuss.
The exact inclusion and exclusion criteria are still missing. I recommend to include a table presenting this information
The formal aspects are still not in the right form. Please adjust the formal points at the end of the present form of the paper following the guidelines of Applied Sciences.
The other section were improved.
The English language is improved. Only minor correstions are needed.
Author Response
Dear Authors
Thank you for this new version of your work. I think the paper is improved. However, there are some corrections still needed.
The title is improved. Furthermore, the abstract is now in an acceptable form.
The introduction is improved. Thank you.
Dear Reviewer, thank you for your valuable feedback and your thoughtful suggestion regarding our narrative review. We genuinely appreciate your engagement with our research.
In the material and methods section: Why did you not use "dental traumatology" or "tooth fractures" as search string. This might lead to a higher number of possible papers for inclusion. Please explain and discuss.
We understand your point about potentially using broader search terms like "dental traumatology" or "tooth fractures," but we would like to clarify our approach and rationale for selecting our search strings: in our study, we aimed to strike a balance between inclusivity and specificity in our search strategy. Our primary objective was to conduct a narrative review of the literature specifically related to sports mouthguard devices. We chose to focus on a more precise set of keywords to ensure that the papers we retrieved were directly related to our main point.
Acknowledging that different research questions may benefit from distinct search strategies, we appreciate your perspective on this matter. However, in this case, we believe that our chosen search strategy was well-suited to our research objectives and allowed us to maintain the focus and relevance necessary for our study.
The exact inclusion and exclusion criteria are still missing. I recommend to include a table presenting this information
The inclusion and exclusion criteria can be found in section 2. Narrative Literature Review Search Strategy, lines 97-103, as follows: “Original research and review articles in the English language were only included in this review. A total of 920 articles were found and 39 articles were selected and included in this review. The terms used for the search were “sports mouthguard”, “mouthguards and orofacial traumas”, “mouthguards and additive manufacturing”. Further searches were conducted in the references list of the relevant articles dealing with the topic of interest. Editorials, Letters to the Editor, Case Reports, and short communication were excluded.”
The formal aspects are still not in the right form. Please adjust the formal points at the end of the present form of the paper following the guidelines of Applied Sciences.
In response to your comment about the formal aspects of the paper, we understand your concern and we made the necessary adjustments to align with the guidelines of Applied Sciences. Your input is valuable to us, and we are committed to enhancing the paper's formal presentation while maintaining the integrity of our research.
The other section were improved.
Thank you.
Round 3
Reviewer 1 Report
Thank you, for the interesting feedback.
None.
Author Response
Thank you for your time and effort in evaluating the present study.